# Few-Shot Task Learning through Inverse Generative Modeling

**Aviv Netanyahu**[1][*], **Yilun Du**[1,2], **Antonia Bronars**[1], **Jyothish Pari**[1], **Joshua Tenenbaum**[1], **Tianmin Shu**[3], and **Pulkit Agrawal**[1]

[1]Massachusetts Institute of Technology
[2]Harvard University
[3]Johns Hopkins University

## Abstract

Learning the intents of an agent, defined by its goals or motion style, is often extremely challenging from just a few examples. We refer to this problem as task concept learning and present our approach, Few-Shot Task Learning through Inverse Generative Modeling (FTL-IGM), which learns new task concepts by leveraging invertible neural generative models. The core idea is to pretrain a generative model on a set of basic concepts and their demonstrations. Then, given a few demonstrations of a new concept (such as a new goal or a new action), our method learns the underlying concepts through backpropagation without updating the model weights, thanks to the invertibility of the generative model. We evaluate our method in five domains – object rearrangement, goal-oriented navigation, motion caption of human actions, autonomous driving, and real-world table-top manipulation. Our experimental results demonstrate that via the pretrained generative model, we successfully learn novel concepts and generate agent plans or motion corresponding to these concepts in (1) unseen environments and (2) in composition with training concepts.

## 1 Introduction

The ability to learn concepts about a novel task, such as the goal and motion plans, from a few demonstrations is a crucial building block for intelligent agents – it allows an agent to learn to perform new tasks from other agents (including humans) from little data. Humans, even from a young age, can learn various new tasks from little data and generalize what they learned to perform these tasks in new situations [1].

In machine learning and robotics, this class of problems is referred to as Few-Shot Learning [2]. Despite being a widely studied problem, it remains unclear how we can enable machine learning models to learn concepts of a novel task from only a few demonstrations and generalize the concepts to new situations, just like humans do. Common approaches learn policies either directly, which often suffer from covariate shift [3], or via rewards [4–6], which are largely limited to previously seen behavior [7]. In a different vein, other work has relied on pretraining on task families and assumes that task learning corresponds to learning similar tasks to ones already seen in the task family [8, 9].

Inspired by the success of generative modeling in few-shot visual concept learning [10–12], where concepts are latent representations, in this work, we investigate whether and how few-shot task concept learning can benefit from generative modeling as well. Learning concepts from sequential demonstrations rather than images is by nature more challenging due to sequential data often not

---

[*]Correspondence to Aviv Netanyahu <avivn@mit.edu>. Project website https://avivne.github.io/ftl-igm.

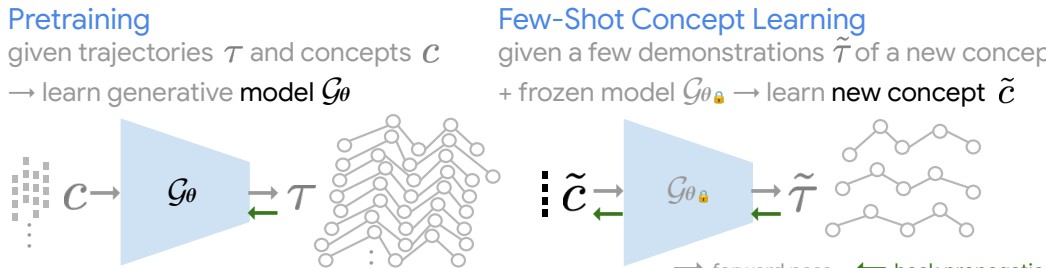

Figure 1: **Few-shot concept learning.** Given paired task demonstration $\tau$ (*e.g.*, 'walk') and concept $c$ (a latent representation of the task), we train a generative model $\mathcal{G}_\theta$ to generate behavior from a concept. Then, given demonstrations of a new behavior $\tilde{\tau}$ (*e.g.*, 'jumping jacks') without its concept label, we aim to learn its concept representation by optimizing concept $\tilde{c}$ as input to frozen $\mathcal{G}_\theta$.

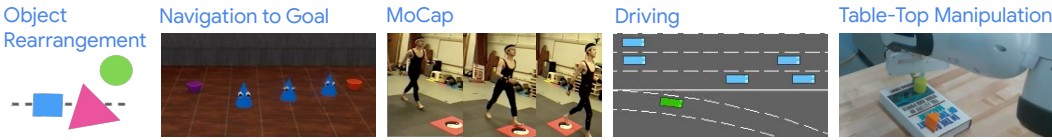

Figure 2: **Experiment Domains.** We extensively evaluate our approach for various domains.

satisfying the i.i.d. assumption in machine learning [13]. In particular, we assume access to a large pretraining dataset of paired behaviors and task representations to learn a conditional generative model that synthesizes trajectories conditioned on task descriptions. We hypothesize that by learning a generative model conditioned on explicit representations of behavior, we can acquire strong priors about the nature of behaviors in these domains, enabling us to more effectively learn new behavior that is not within the pretraining distribution, given a limited number of demonstrations, and further generate the learned behavior in new settings.

To this end, we propose Few-Shot Task Learning through Inverse Generative Modeling (FTL-IGM). In our approach, we first pretrain a large conditional generative model which synthesizes different trajectories conditioned on different task descriptions. To learn new tasks from a limited number of demonstrations, we then formulate few-shot task learning as an *inverse generative modeling problem*, where we find the latent task description, which we refer to as a *concept*, which maximizes the likelihood of generating the demonstrations. This approach allows us to leverage the powerful task priors learned by the generative model to learn the shared concepts between demonstrations without finetuning the model (Figure 1). We demonstrate this approach in various domains: object rearrangement, where concepts are relations between objects, goal-oriented navigation, where concepts are target attributes, motion capture, where concepts are human actions, autonomous driving, where concepts are driving scenarios, and real-world table-top manipulation where concepts are manipulation tasks (Figure 2).

New concepts are either (1) compositions of training concepts (*e.g.*, multiple desired relations between objects that define a new object rearrangement concept) or (2) new concepts that are not explicit compositions in the natural language symbolic space of training concepts (*e.g.*, a new human motion 'jumping jacks' is not an explicit composition of training concepts 'walk', 'golf' etc.) Thanks to generative models' compositional properties that enable compositional concept learning [14], in addition to being able to learn a single concept from demonstrations directly, FTL-IGM learns compositions of concepts from demonstrations that, when combined, describe the new concept.

We show that our approach generates diverse trajectories encapsulating the learned concept. We achieve this due to two properties of generative models. First, these models have shown strong interpolation abilities [15, 16], which allow generating the new concept on new initial states they were not demonstrated from. Second, these models have compositional properties that enable compositional trajectory generation [17], which allow composing learned concepts with training concepts to synthesize novel behavior that was not demonstrated (*e.g.*, 'jumping jacks' and 'walk'), see Figure 3. We further demonstrate that our approach addresses a unique challenge introduced in learning task concepts: we utilize plans generated by learned concepts in a closed-loop fashion.

Our main contributions are (1) formulating the problem of task learning from few demonstrations as Few-Shot Task Learning through Inverse Generative Modeling (FTL-IGM), (2) adapting a method for efficient concept learning to this problem based on the new formulation, and (3) a systematic evaluation revealing the ability of our method to learn new concepts across a diverse set of domains.

## 2 Related Work

**Learning from few demonstrations.** Our problem setting is closely related to learning from few demonstrations. There has been much work on learning to generate agent behavior given few demonstrations. There are several common approaches to this problem. First, behavior cloning (**BC**) is a supervised learning method to learn a policy from demonstrations that predicts actions from states. Similar to our framework, goal-conditioned BC can predict states from task representations and states [18]. Finetuning these models to learn new behaviors requires labeled demonstrations of the new task. We assume unlabeled demonstrations. BC often suffers from covariate shift [3] and fails to generate the demonstrated behavior in novel scenarios. This can be mitigated by assuming access to a human in the loop [19]. Second, the inverse reinforcement learning (**IRL**) framework learns a policy that maximizes the return of an explicitly [5, 6, 20] or implicitly [21] learned reward function. These works learn a reward for a single task or for a set of tasks (*e.g.*, goal-conditioned IRL [22, 23] and multi-task IRL [24]). While IRL is more data efficient than BC, it is computationally costly due to learning a policy every iteration via an inner reinforcement learning (RL) loop. Additionally, it requires access to taking actions in the environment during training and when faced with a new task, we have to retrain the reward again. A third approach is **inverse planning** [25–27], which can robustly infer concepts such as goals and beliefs even in unseen scenarios. However, it assumes access to a planner, knowledge about environment dynamics, and the task/goal space. Finally, **in-context** learning approaches [8, 28, 29] learn actions in a supervised manner by representing the task with demonstrations. This allows few-shot generalization without further training.

In contrast, we do not learn an action-generating policy directly or via a reward function. For concept learning, we do not assume having access to any given planner, world model, actions, rewards, or prior over the task space. Instead, we learn *concepts* (task representations) from demonstrations via a pretrained generative model that takes a concept as input and directly produces state sequences. We then input the learned concept into the generative model to produce behavior similar yet diverse to the demonstrated one. We further demonstrate how to use these state sequences with a planner to take actions and achieve the desired behavior. The idea of concept learning via generative models has been explored for computer vision applications [11, 12]. We build on this work and show how to extend it to learn agent task concepts. Our work also differs from prior works on learning trajectory representations [30–34]. These works focus on learning plans over trajectory embeddings, whereas we learn a task representation from demonstrations on which we condition to generate behavior.

**Generative Models in Decision Making.** There has been work on generative modeling for decision-making, including generative models for single-agent behaviors, such as implicit BC [35], Diffuser [36, 17], Diffusion Policy [37], Decision Transformer [38–40], and for multi-agent motion prediction such as Jiang et al. [41]. The success of diffusion policy in predicting sequences of future actions has led to 3D extensions [42], and combined with ongoing robotic data collection efforts [43] and advanced vision and language models, has led to vision-language-action generative models [44–46]. In this work, we utilize a conditional generative model for the *inverse* problem, *i.e.*, learning concepts from demonstrations.

**Composable representations.** There has been work on obtaining composable data representations. $\beta$-VAE [47] learns unsupervised disentangled representations for images. MONet [48] and IODINE [49] decompose visual scenes via segmentation masks and COMET [50] and [12] via energy functions. There is also work on composing representations to generate data with composed concepts. Generative models can be composed together to generate visual concepts [14, 51–55] and robotic skills [17]. The generative process can also be altered to generate compositions of visual [56–59] and molecular [60] concepts. We aim to obtain task concepts and generate them in composition with other task concepts.

## 3 Formulation

Inspired by recent success in large generative models, we propose a generative formulation for learning specific behavior given a small set of demonstrations, which we term Few-Shot Task Learning through

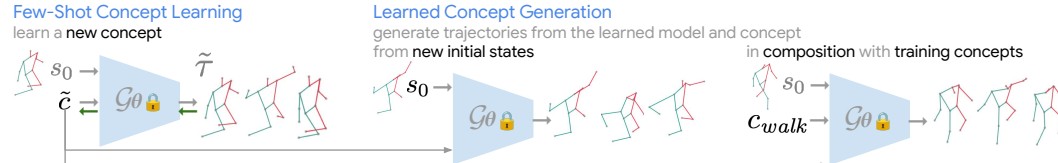

Figure 3: **Diverse learned concept generation.** We generate versions of the new behavior conditioned on the learned concept and (1) new initial states and (2) composed with other concepts.

Inverse Generative Modeling (**FTL-IGM**). In our formulation, we assume access to a large pretraining dataset $D_{\text{pretrain}} = \{(\tau_i, c_i)\}_{i=1}^N$ of *state*-based sequences $\tau_i = \{s_0, s_1, ...\} \subseteq \mathcal{T}$ of states from state space $S$ annotated with meta-data "concepts" $c_i \in \mathcal{C} \subseteq \mathbb{R}^n$ describing trajectories. This assumption is often not prohibitive in practice. There is typically a vast amount of existing data collected from the internet or prior exploration in an environment, which may only need to be weakly annotated to characterize the trajectory, *e.g.*, the goal state. Given $D_{\text{pretrain}}$, we learn a conditional generative model $\mathcal{G}_\theta : \mathcal{C} \times \mathcal{S} \to \mathcal{T}$ conditioned on concepts and initial states, which learns to generate future trajectories. We train the parameters of $\mathcal{G}_\theta$ to maximize likelihood $\arg\max_\theta \mathbb{E}_{\tau, c \sim D_{\text{pretrain}}}[\log \mathcal{G}_\theta(\tau|c, s_0)]$.

Then, given an unlabeled demonstration dataset $D_{\text{new}} \sim \mathcal{D}$, we formulate learning a new concept $\tilde{c}$ that is used to sample trajectories from $\mathcal{D}$ as *inverting* the generative model. In particular, we learn new concept $\tilde{c}$ so that our frozen conditional generative model $\mathcal{G}_\theta$ maximizes the likelihood of trajectories in $D_{\text{new}}$, corresponding to $\arg\max_{\tilde{c}} \mathbb{E}_{\tau \sim D_{\text{new}}}[\log \mathcal{G}_\theta(\tau|\tilde{c}, s_0)]$. We find that this design choice enables us to leverage the priors learned by $\mathcal{G}_\theta$ from $D_{\text{pretrain}}$ to effectively learn concepts from $D_{\text{new}}$ given very few demonstrations, even if the demonstrated $D_{\text{new}}$ deviates from the concept labels $c$ seen in $D_{\text{pretrain}}$. For evaluation in closed loop, we further assume access to a planner that given two states plans which action to take in the environment, sometimes via access to simulation in the environment. We use this planner sequentially to make decisions in the environment.

The key difference between our approach to few-shot adaptation from demonstrations and prior approaches is the assumption and usage of a large pretraining dataset of paired behaviors and concepts $D_{\text{pretrain}}$ combined with an invertible generative model. We learn new concepts solely from demonstrations without finetuning model weights or taking actions in the environment by relying on the pretrained concept space.

## 4   Few-Shot Concept Learning Based on FTL-IGM

We adapt a few-shot concept learning method to task concepts based on the FTL-IGM framework. During training we learn a generative model $\mathcal{G}_\theta$ from training $\{(\tau_i, c_i)\}_i$ pairs. We then freeze $\mathcal{G}_\theta$, and given demonstrations of a new task $\{\tilde{\tau}\}_i$, optimize a concept $\tilde{c}$ to produce the new behavior via $\mathcal{G}_\theta$. We then generate a diverse set of behaviors via $\mathcal{G}_\theta$, either for the learned concept $\tilde{c}$ conditioned on new initial states or for compositions of $\tilde{c}$ with other concepts.

### 4.1   Training a diffusion model to generate behavior

A **diffusion model** is a generative model that given a forward noise adding process $q(x_t|x_{t-1}) := \mathcal{N}(x_t; \sqrt{1 - \beta_t}x_{t-1}, \beta_t \mathbf{I})$ starting from data $x_0$ according to a variance schedule $\beta_1, ..., \beta_T$, learns the reverse process $p_\theta(x_{t-1}|x_t) := \mathcal{N}(x_{t-1}; \mu_\theta(x_t, t), \Sigma_\theta(x_t, t))$. Ho et al. [61] simplify the training objective to estimate noise $\mathbb{E}_{t \sim \mathcal{U}\{1,T\}, x_0, \epsilon \sim \mathcal{N}(0,\mathbf{I})}[||\epsilon - \epsilon_\theta(x_t, t)||^2]$ where $x_t$ is produced by adding noise $\epsilon$ to data $x_0$ by the forward noising process at diffusion step $t$, $q(x_t|x_0) := \mathcal{N}(x_t; \sqrt{\bar{\alpha}_t}x_0, (1 - \bar{\alpha}_t)\mathbf{I})$ where $\bar{\alpha}_t := \Pi_{s=1}^t(1 - \beta_s)$. Dhariwal and Nichol [62] enable **conditioned generation** by guiding the reverse noising process with classifier gradients. The noise prediction becomes $\hat{\epsilon} = \epsilon_\theta(x_t, t) - \omega\sqrt{1 - \bar{\alpha}_t}\nabla_{x_t} \log p_\phi(y|x_t, t)$ where classifier $p_\phi(y|x_t, t)$ is trained on noisy images, and $\omega$ is the guidance scale. Ho and Salimans [63] introduce **classifier-free guidance** that achieves the same objective without the need for training a separate classifier. This is done by learning a conditional and unconditional model by removing the conditioning information with dropout during training. The noise prediction is then $\hat{\epsilon} = \epsilon_\theta(x_t, t) + \omega(\epsilon_\theta(x_t, y, t) - \epsilon_\theta(x_t, t))$. Ramesh et al. [64] and Nichol et al. [65] demonstrate how this idea can be used to generate images conditioned on a class. Diffusion models have recently shown success as generative models for **decision making**

[36, 17]. Specifically, Ajay et al. [17] used a conditional classifier-free guidance diffusion model [63] to generate trajectories of future states to reach given an input observation. We adopt this objective and learn a denoising model $\epsilon_\theta$ conditioned on latent concepts and initial observed states to estimate noise of a future state trajectory:

$$\mathbb{E}_{(\tau,c)\sim D_{\text{pretrain}},\epsilon\sim\mathcal{N}(0,\mathbf{I}),t\sim\mathcal{U}\{1,T\},\gamma\sim\text{Bern}(p)}[||\epsilon - \epsilon_\theta(x_t(\tau),(1-\gamma)c + \gamma c_\emptyset, s_0, t)||^2] \quad (1)$$

where $p$ is the probability of removing conditioning information which is then replaced by dummy condition $c_\emptyset$, and $s_0$ is the initial state corresponding to trajectory $\tau$. $x_t(\tau)$ is obtained from $x_0 = \tau$ by the forward noising process. We then extend this approach for the inverse problem, namely, learning a concept from demonstrations.

## 4.2 Few-shot concept learning

Gal et al. [11] use a frozen generative model to **learn visual concept** representations from few images depicting the concept by optimizing the model's input $v_* = \arg\min_v \mathbb{E}_{x_0,v,\epsilon\sim\mathcal{N}(0,1),t}\left[||\epsilon - \epsilon_\theta(x_t, c_\theta(v), t)||_2^2\right]$, where $c_\theta$ and $\epsilon_\theta$ are fixed. Liu et al. [12] extend this and **learn visual concept compositions** with a pretrained diffusion model in an unsupervised manner. Namely, from a set of images that depict various concepts, for each image $x^i$ they learn a set of weights $\omega_k^i$ and a shared set of visual concepts for all images $c_k$, $\hat{\epsilon} = \epsilon(x_t^i, t) + \sum_{k=1}^{K} \omega_k^i(\epsilon(x_t^i, c_k, t) - \epsilon(x_t^i, t))$. We extend these formulations to inferring multiple concepts, whose composition describes a single task concept, from few demonstrations of a task.

Given a trained diffusion model $\epsilon_\theta$ and demonstrations of a new concept $\{\tilde{\tau}\}_i$ from $D_{\text{new}}$, we learn concepts $\{\tilde{c}_1, ..., \tilde{c}_K\}$ for $K \geq 1$ and their weights $\{\omega_1, ..., \omega_K\}$ that best describe the demonstrations. Starting from uniformly sampled concept embeddings $\tilde{c}_k \sim \mathcal{U}([0,1]^n)$, we freeze $\epsilon_\theta$, and optimize $\tilde{c}_k$ and $\omega_k$:

$$\mathbb{E}_{\epsilon\sim\mathcal{N}(0,\mathbf{I})}[||\epsilon - (\epsilon_\theta(x_t(\tilde{\tau}), c_\emptyset, s_0, t) + \sum_{k=1}^{K} \omega_k(\epsilon_\theta(x_t(\tilde{\tau}), \tilde{c}_k, s_0, t) - \epsilon_\theta(x_t(\tilde{\tau}), c_\emptyset, s_0, t)))||^2]. \quad (2)$$

We find that this compositional approach enables us to effectively represent and learn new demonstrations, even when demonstrations are substantially different than those seen in training tasks.

## 4.3 Generating the learned concept

After learning concepts $\tilde{c}_k$, whose composition describes the new task $\tilde{c}$, we evaluate the behavior it generates by initializing $x_T(\tau) \sim \mathcal{N}(0, \alpha\mathbf{I})$, and compute $x_t \sim \mathcal{N}(\mu_{t-1}, \alpha\Sigma_{t-1})$ iteratively as a function of the estimated denoising function $\hat{\epsilon}(\epsilon_\theta)$, where $\mu$ and $\Sigma$ are the mean and variance that define the reverse process, and $\alpha \in [0, 1)$ is a scaling factor that leads to lower temperature samples, until generating $x_0 = \tau$ representing the trajectory of the agent. The denoising function is constructed by fixed or learned weights as defined in Eq. 2 and by any number of concepts $\geq 1$. The applications of the generation procedure can be summarized as:

**Learned concept and demonstrated initial states.** We apply our learned concept to a set of demonstrated initial states. In domains where the initial state and concept jointly determine optimal behavior, the generated trajectory corresponds to optimal actions to execute (*e.g.* goal-oriented navigation). In contrast to other domains where the initial state is irrelevant for a task due to the randomness in sampling $x_T(\tau)$ (*e.g.* motion capture), generated trajectories correspond to diverse plausible behaviors exhibiting the learned concept.

**Learned concept and novel initial states.** We further apply the generation procedure from our learned concept on novel initial states, to generate trajectories of new behaviors exhibiting our conditioned concept. Prior methods may suffer from covariate shift in this setting [21]. We empirically show that our method is less prone to this problem.

**Learned concept composed with other concepts.** Finally, we modify the generation procedure of our newly learned concept to generate trajectories that simultaneously exhibit other concepts. To generate a trajectory with an added another concept, we add another term to the sum in Eq. 2 where the learned concept is composed with a training concept $c_k$ and its weight $\omega_k$: $\omega_k(\epsilon_\theta(x_t(\tau), c_k, s_0, t) - \epsilon_\theta(x_t(\tau), c_\emptyset, s_0, t))$. This modified generation procedure constructs trajectories which exhibits behavior that has a composition of the learned concept and the other specified concepts [14].

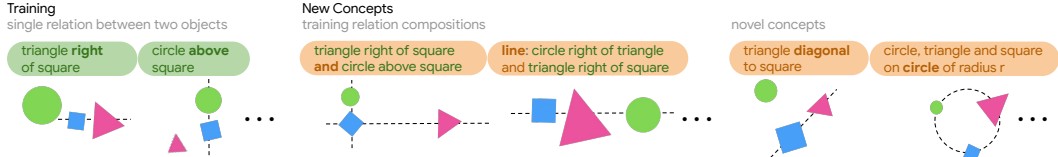

Figure 4: **Object rearrangement.** Training concepts are single pairwise relations ('A right of/above B'), and new concepts are either compositions of training concepts ('A right of/above B' ∧ 'B right of/above C') or new relations ('A diagonal to B', 'A, B, C on circle circumference of radius r').

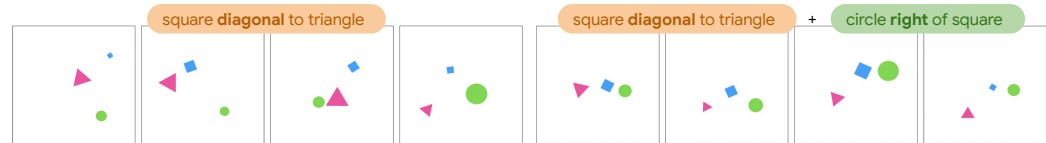

Figure 5: **Object rearrangement new concept qualitative evaluation.** Learning the new concept 'square diagonal to triangle' and composing it with the training concept 'circle right of square'.

Similarly to Ajay et al. [17], in environments where an inverse dynamics model is provided, we generate trajectories in a closed loop. We execute actions calculated by the inverse dynamics given the predicted plan by the model and then repeatedly replan given new observations.

## 5  Experiments

We demonstrate results in four domains where concept representations are T5 [66] embeddings of task descriptions in natural language for training, and empty string embeddings for the dummy condition. During few-shot concept learning, we are provided with three to five demonstrations of a composition of training concepts or of a novel concept that is not an explicit composition of training tasks in natural language symbolic space. We ask a model to learn the concept from these demonstrations.

### 5.1  Task-Concepts

**Learning concepts describing goals that are spatial relations between objects.** Object rearrangement is a common task in robotics [67–69] and embodied artificial intelligence (AI) [70, 71], serving as a foundation for a broader range of tasks such as housekeeping and manufacturing. Here, we use a 2D object rearrangement domain to evaluate the ability of our method to learn task specification concepts. Given a concept representing a relation between objects, we generate a single state describing that relation. The concept in a training example describes the relation (either 'right of' or 'above') between only one pair of objects (out of three objects) in the environment. Then, a model must learn compositions of these pairwise relations and new concepts such as 'diagonal' and 'circle' (see Figure 4). The results in Figures 5 and 6 demonstrate that our method learns unseen compositions of training concepts and new concepts. They further demonstrate how our method composes new concepts with learned concepts. For additional qualitative results, please refer to Appendix A.

While successful in most cases, there are also a few failure examples. The accuracy for the new 'circle' concept is low (0.44) compared to the mean over task types in Figure 6 Object Rearrangement New Concept (0.82 ± 0.09). This is most likely due to this concept lying far out of the training distribution. The task 'square right of circle ∧ triangle above circle' has low accuracy for 2 concepts (0.32) compared to the mean in Table 2 Object Rearrangement Training Composition (0.75 ± 0.11). This may arise from the combined concept-weight optimization process – as there is no explicit regularization on weights, they may converge to 0 or diverge. In Figure 12, we show that concept components may or may not capture new concept relations.

**Learning concepts describing goals based on attributes of target objects.** We test our method in a goal-oriented navigation domain adapted from the AGENT dataset [72], where an agent navigates to one of two potential targets. Conditioned on a concept representing the attributes of the desired

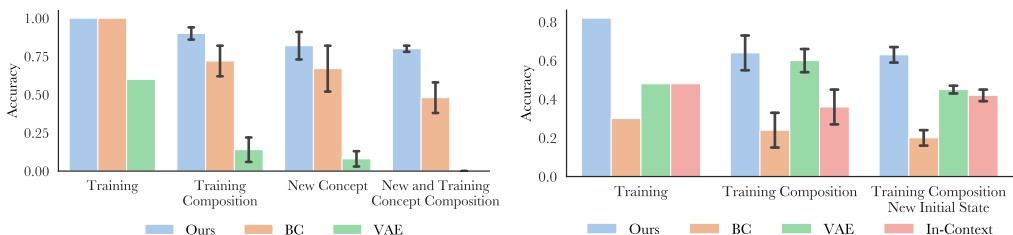

Figure 6: **Object rearrangement (left) and AGENT closed loop (right) quantitative evaluation on training and few-shot novel concept learning.** Accuracy of FTL-IGM (ours), BC, VAE, and In-Context over concept generation of training concepts, novel compositions, novel concepts, and new initial states. We plot the average and standard error over new task types. Full details of the evaluation metrics appear in Appendix B and for baselines implementation in Appendix C.

target object and initial state, we generate a state-based trajectory describing an agent navigating to the target. Each object has a color and a shape out of four possible colors and four shapes. During training, we provide 16 target-distractor combinations that include all colors and shapes (but not all combinations), and a concept is conditioned on one of the target's attributes (*e.g.*, color). We introduce new concepts defined by both target attributes, including (1) unseen color-shape target combinations and (2) new target-distractor combinations. Figure 7 shows an example. In training, we see bowl and red object targets. A new concept includes a novel composition as the target – red bowl. The new concept distractor objects (green bowl and red sphere) were introduced during training, but they were not paired with a red bowl as the target. As Figure 13 shows, our method successfully learns concepts where targets are new compositions of target attributes in settings with new target-distractor pairs and generalizes to new initial object states. We further evaluate our model and baselines in closed loop (Figure 6) by making an additional assumption that a planner is provided. The planner produces an action given a current state and a future desired state predicted by a model.

**Learning concepts describing human motion.** Unlike prior work on learning to compose human poses from motion capture (MoCap) data [e.g., 73, 74], here we focus on the inverse problem – learning new actions from MoCap data. In particular, we use the CMU Graphics Lab Motion Capture Database (http://mocap.cs.cmu.edu/). We train on various human actions in the database and few-shot learn three novel concepts (see Appendix B.3 for details). Learning tasks from few demonstrations is especially beneficial in this domain since describing motion concepts in words could be hard. In Table 5.1, we ask five human volunteers to select generated behaviors that depict training and new concepts. We demonstrate quantitatively that our method generates human motion that captures the desired behavior across training and new behaviors. We qualitatively demonstrate how our method generates learned new concepts ('jumping jacks' and 'breaststroke') from new initial states and composes 'jumping jacks' with training concepts 'walk', 'jump', and 'march'.

Results are best viewed on our website website. We compare motion generated by our method to various baselines on new initial states for 'jumping jacks' and 'breaststroke'. Our method is noisy yet captures the widest range of motion. While other methods often produce smoother trajectories, they mostly capture local (VAE) or degenerate (BC, In-Context, Language) motion.

**Learning concepts describing driving scenarios.** In an Autonomous Driving domain [75], an agent acts in a challenging multi-agent environment to complete a driving task. We train on several driving scenarios ('highway', 'exit', 'merge', and 'intersection') and learn a new driving scenario ('roundabout') from several demonstrations (see Figure 8 and further details in Appendix B.4). We evaluate this scenario in closed loop on new initial states, assuming access to a planner that can simulate taking actions in the environment. Over two evaluation metrics (crash and task completion rate), our method achieves overall best results (Figure 9).

**Learning concepts describing real-world table-top manipulation tasks.** We evaluate our method's capability to learn a novel concept for real-world table-top manipulation with a Franka Research 3 robot. Training concepts include 'pick green circle and place on book', 'pick green circle and place on elevated white surface', 'push green circle to orange triangle' and 'push green circle to orange triangle around purple bowl'. The new scenario includes pushing the green circle to the orange triangle on a book (Figure 10). We evaluate training pushing in closed loop and and achieve success

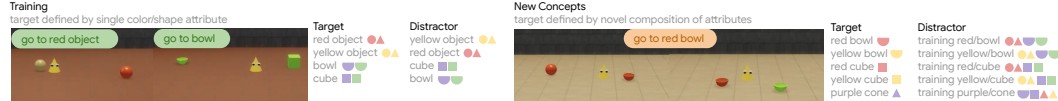

Figure 7: **Goal-oriented navigation.** In training, targets are defined by a single attribute (color or shape), and in new concepts, targets are defined by a novel combination of color and shape attributes. To make the setting more challenging, distractor objects in new concept demonstrations have a combination of attributes that are within the training distribution.

| Setting | BC | VAE | Ours |
|---|---|---|---|
| Training | $20 \pm 16.3$ | $0 \pm 0$ | $\mathbf{80 \pm 16.3}$ |
|  | $66.7 \pm 0$ | $46.7 \pm 16.3$ | $\mathbf{100 \pm 0}$ |
| New Concept | $26.7 \pm 13.3$ | $0 \pm 0$ | $\mathbf{73.3 \pm 13.3}$ |
| New Initial State | $40 \pm 24.9$ | $26.7 \pm 13.3$ | $\mathbf{80 \pm 16.3}$ |

Table 1: **MoCap human experiment.** For training concepts and new concepts on new initial states, we report (top) percentage of time each method is the most successful at depicting a concept and (bottom) percentage of time each method depicts a concept. Mean and standard deviation are calculated over the number of scenarios in each setting and human subjects.

(90% accuracy). We evaluate the new concept, elevated pushing, against a baseline that conditions on the training 'push green circle to orange triangle' representation in the new scenario setup where the objects are placed on a book. Learning the new concept succeeds (55% accuracy) while using the training representation fails (15% accuracy) and the robot often pushes the book instead of the object. Details are in Appendix B.5 and qualitative results are on our website.

## 5.2 Baselines

**Goal conditioned behavior cloning.** We compare our method with goal-conditioned behavior cloning (BC), which, given a condition and a state, outputs the next state in a sequence. It is trained on our paired pretraining dataset and learns concepts by optimizing the input condition to maximize the likelihood of new concept demonstration transitions. We test its ability to compose new learned concepts with training concepts naively by adding conditions that are then inputted into the model. We observe that even though goal-conditioned BC has access to the pretrained dataset and conditions, and while it may learn new concepts, it suffers from covariate shift on new initial states and lacks the ability to generalize to novel compositions of the learned concept with training concepts (Figures 6, 9, 13). To achieve these generalization capacities, we need a model that can process interpolated (initial states) and composed (concepts) conditions, such as our generative model.

**Learning the concept space with a VAE.** We compare our method with VAE [76] that does not utilize the concepts in the paired pretraining data but rather learns the concept space by reconstructing pretraining data trajectories through their encoded representation $z$. Trajectories are generated via a decoder conditioned on an initial state and $z$ with added noise. $z$ is obtained by encoding a trajectory for training evaluation and by fixing the decoder and optimizing $z$ to generate a given demonstration when learning a new concept. We find that the VAE model learns a latent space that captures training and new concepts but does not enable generalization to new initial states (Figures 6, 9, 13). This highlights the importance of concept representations in the pretrained data.

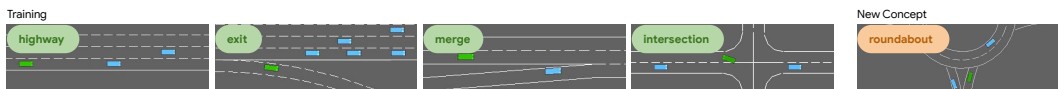

Figure 8: **Autonomous driving.** A controlled agent (green) completes various driving objectives as fast as possible while sharing the road with other vehicles (blue) and avoiding collisions. Training concepts: 'highway', 'exit', 'merge', and 'intersection', new concept: 'roundabout'.

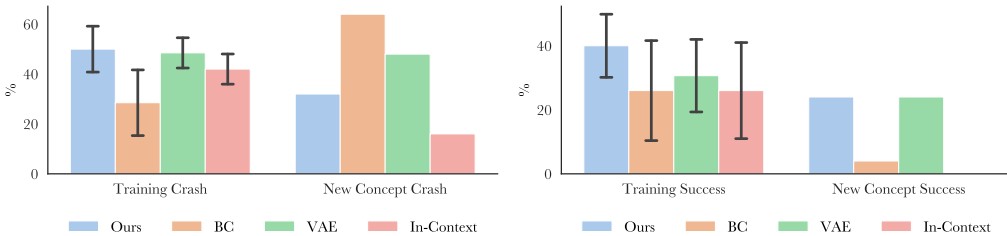

Figure 9: **Driving crash (left) and success (right) rates.** Crash rate (lower is better) and task completion rate (higher is better) averaged over training tasks. We report standard errors over training tasks and accuracy over 50 trajectories generated from the learned new concept. VAE has a high completion rate yet a high crash rate. In-context has a low crash rate yet 0% success rate – typically, the controlled vehicle reaches the roundabout's center but does not complete the crossing. Overall, our method learns to complete the roundabout crossing with competitive crash and success rates.

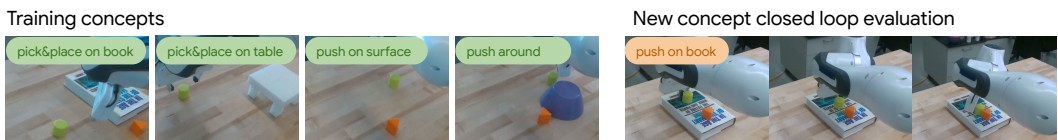

Figure 10: **Table-top manipulation.** Training concepts: pick-and-place onto elevated surfaces and table-top pushing. New concept: pushing on an elevated surface.

**In-Context learning.** We compare our approach with training a method to in-context learn from demonstrations. Specifically, we compare our approach to Xu et al. [28] using the pretrained dataset for few-shot behavior generation without further training. We sequentially predict states from demonstrations of a concept and window of current states and show that our method adapts better to new concepts (Figures 6, 9, 13). This emphasizes the need to learn explicit concept representations.

**Conditioning on Language Descriptions of New Concepts.** There has been work on generating actions from language instructions [77–79]. We demonstrate that in our setup, merely providing new concept language instructions embedded with T5 (as in our pretraining dataset) is insufficient, and generalization is better when learning concepts from few demonstrations. For AGENT, training compositions on new initial states has an average accuracy and standard error of $0.63 \pm 0.07$, lower than ours ($0.73 \pm 0.07$). For Object Rearrangement, training compositions ($0.2 \pm 0.07$), new concept ($0.2 \pm 0.05$), and new and training concept compositions ($0.13 \pm 0.04$) accuracies are significantly lower than ours ($0.9 \pm 0.04$, $0.82 \pm 0.09$ and $0.8 \pm 0.02$). For MoCap, we demonstrate qualitatively that instead of capturing new human actions, the agent transitions into walking. Results are best viewed on our website.

## 5.3 Learning two concepts yields higher accuracy than one concept

When learning weights together with concepts, we check the effect of the number of learned concepts and weights. We report results in Table 2 for Object Rearrangement, AGENT, and Driving, and find that, on average, learning two concepts improves concept learning. We demonstrate qualitatively for MoCap that learning two conditions is preferable. In 'jumping jacks', we observe that the motion lacks raising and lowering both arms and in 'breaststroke', it lacks complete arm and upper torso movement. Results are best viewed on our website.

## 5.4 How are learned new concepts related to training concepts?

**New concepts that are compositions of training concepts.** We analyze what the learned two concepts in Object Rearrangement and AGENT learn for novel concept compositions (*e.g.*, 'red bowl'). For each concept (*e.g.*, 'red' and 'bowl'), we generate two sets of 50 samples from the learned components. Table 3 shows accuracy for these sets over the concepts. In some cases (most notably the 'line' concept in Object Rearrangement, 'circle right of triangle and triangle right of square'), each

| Environment | Setting | 1 Concept | 2 Concepts |
|---|---|---|---|
| Object Rearrangement | Training Composition | 0.28±0.11 | 0.75±0.11 |
| | New Concept | 0.49±0.11 | 0.77±0.09 |
| | New+Training | 0.38±0.04 | 0.73±0.04 |
| AGENT | Training Composition | 0.52±0.08 | 0.68±0.13 |
| | New Initial State | 0.54±0.07 | 0.67±0.05 |
| Driving | New Initial State | 0.14 | 0.24 |

Table 2: **Ablation on the number of learned concepts.** We test the effect of the number of learned concepts and weights in FTL-IGM on the generation accuracy of new learned concepts. On average, learning two concept components and their weights is preferable to learning one concept component and its weight. We report average accuracy and standard error over task types for Object Rearrangement and AGENT. Driving includes a single new concept and we report accuracy only.

learned component captures a single composed concept. In other cases, a single learned component captures both concepts (Figure 12).

**New concepts that are not explicit compositions of training concepts in natural language symbolic space.** In Figure 11 we visualize t-SNE [80] embeddings for T5 training concept representations and learned concept component representations. We note that learned components are relatively close to training concepts, maintaining the model's input distribution, yet capture concepts that are not explicit compositions of training concepts.

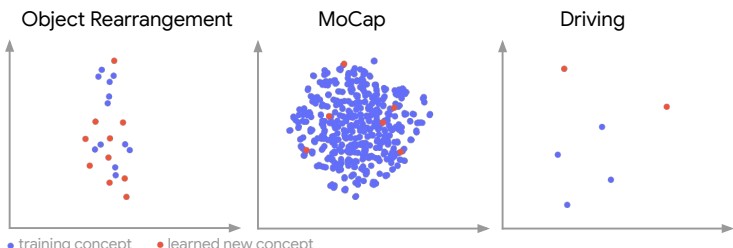

Figure 11: **t-SNE embeddings of new concepts that are not explicit compositions of training concepts.** See interactive version for detailed labels on our website.

## 6  Discussion and Limitations

In this work, we formulate the problem of new task concept learning as Few-Shot Task Learning through Inverse Generative Modeling (FTL-IGM). We adapt a method for concept learning based on this new formulation and evaluate task concept learning against baselines in four domains. Our extensive experimental results show that, unlike the baselines, FTL-IGM successfully learns novel concepts from a few examples and generalizes the learned concepts to unfamiliar scenarios. It also composes learned concepts to form unseen behavior thanks to the compositionality of the generative model. These results demonstrate the efficacy, sample efficiency, and generalizability of FTL-IGM.

However, our work has several limitations. First, while our framework is general for any parameterized generative model, our implementation with a diffusion model incurs high inference time. We note that there is still space for improvement in the MoCap generation quality and in the compatibility rate of demonstrations generated by composing learned and training concepts. In addition, we assume that learned concepts lie within the landscape of training concepts to learn them from a few demonstrations without retraining the model. We have approached the question of what new concepts can be represented by compositions of concepts in this landscape empirically, leaving a theoretical analysis as future work. We are hopeful that with the continued progress in the field of generative AI, more powerful pretrained models will become available. Combined with our framework, this will unlock a stronger ability to learn and generalize various task concepts in complex domains.

## 7 Acknowledgments

This research was also partly sponsored by the United States Air Force Research Laboratory and the United States Air Force Artificial Intelligence Accelerator and was accomplished under Cooperative Agreement Number FA8750-19- 2-1000. The views and conclusions contained in this document are those of the authors and should not be interpreted as representing the official policies, either expressed or implied, of the United States Air Force or the U.S. Government. The U.S. Government is authorized to reproduce and distribute reprints for Government purposes, notwithstanding any copyright notation herein. We acknowledge support from ONR MURI under N00014-22-1-2740 and ARO MURI under W911NF-23-1-0277. Yilun is supported in part by an NSF Graduate Research Fellowship. We would like to thank Abhishek Bhandwaldar for help with the AGENT environment, and Anurag Ajay, Lucy Chai, Andi Peng, Felix Yanwei Wang, Anthony Simeonov and Zhang-Wei Hong for helpful discussions.

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

# A    Additional Results

**Object Rearrangement.**    We display additional states generated by our model for various new concepts. We generate learned concepts that are compositions of training concepts (Figure 14), new concepts that are not explicit compositions of training concepts (Figure 15), and a new concept composed with a training concept (Figure 16). We analyze the learned concepts qualitatively (Figure 12) and quantitatively (Table 3).

**Goal-Oriented Navigation.**    We provide accuracy for open loop evaluation in Figure 13. We further analyze the learned concepts quantitatively (Table 3).

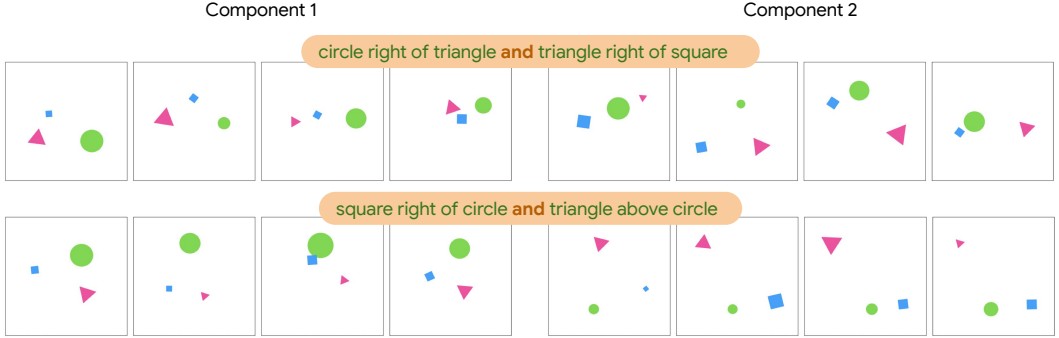

Figure 12: **Object rearrangement new compositions analysis qualitative evaluation.** 'circle right of triangle ∧ triangle right of square' (top) each learned component corresponds to a single composed concept: 'circle right of triangle' (component 1) and 'triangle right of square' (component 2). In 'square right of circle ∧ triangle above circle' (bottom), learned component 2 corresponds to both composed concepts and component 1 to none.

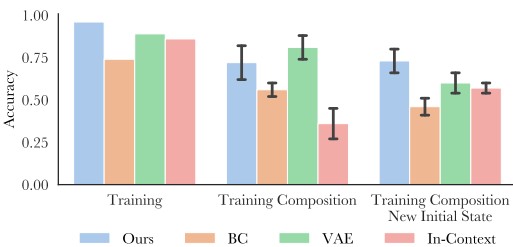

Figure 13: **AGENT open loop evaluation.** We plot the average and standard error over task types.

# B    Data Generation and Evaluation

## B.1    Object Rearrangement

**Training.**    The training dataset of $\sim 11$k samples consists of concepts 'A right of B' and 'A above B', where A and B are one of three objects: circle, triangle, or square. Altogether, there are 12 possible concepts (two relations and three objects where order is important). In the data generation process, A at center position $(x_A, y_A) \in [0, 5]^2$ with radius $r_A \in [0.3, 1]$ and angle $\theta_A \in [0, 2\pi]$ is considered 'right of' B at center position $(x_B, y_B)$ with radius $r_B$ if $x_A > x_B$ and $|y_A - y_B| \leq r_A$. Similarly, A is considered 'above' B if $y_A > y_B$ and $|x_A - x_B| \leq r_A$. We further verify that training objects do not overlap.

**New Tasks.**    The new scenarios include

- Five novel compositions of training concepts (training composition in Figure 6). 'triangle right of square ∧ circle above square', 'square right of triangle ∧ circle above triangle',

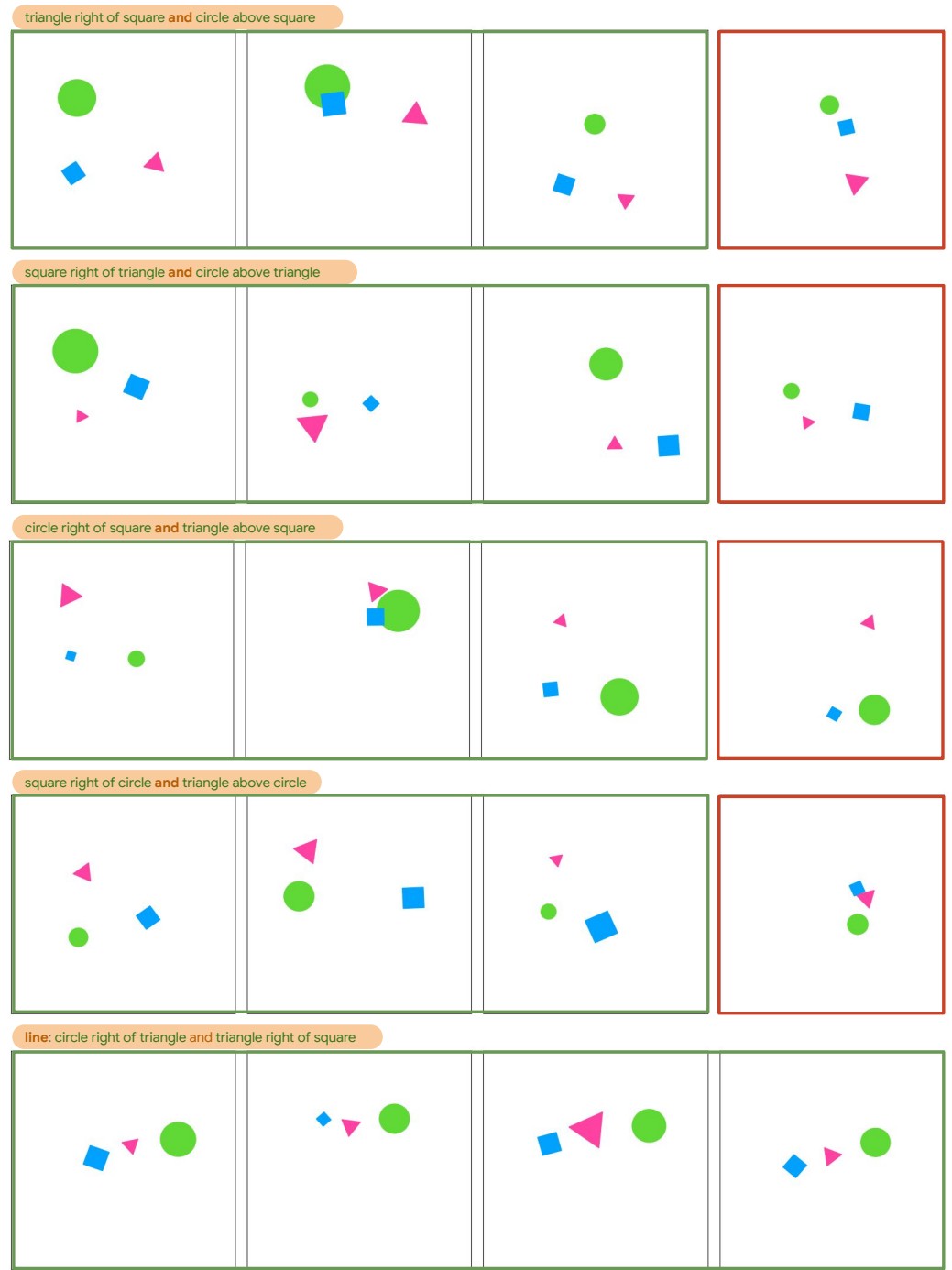

Figure 14: **New concepts that are training concept compositions.** We display successful (green frames) and unsuccessful (red frames) states generated by our model conditioned on learned concepts that are training concept compositions.

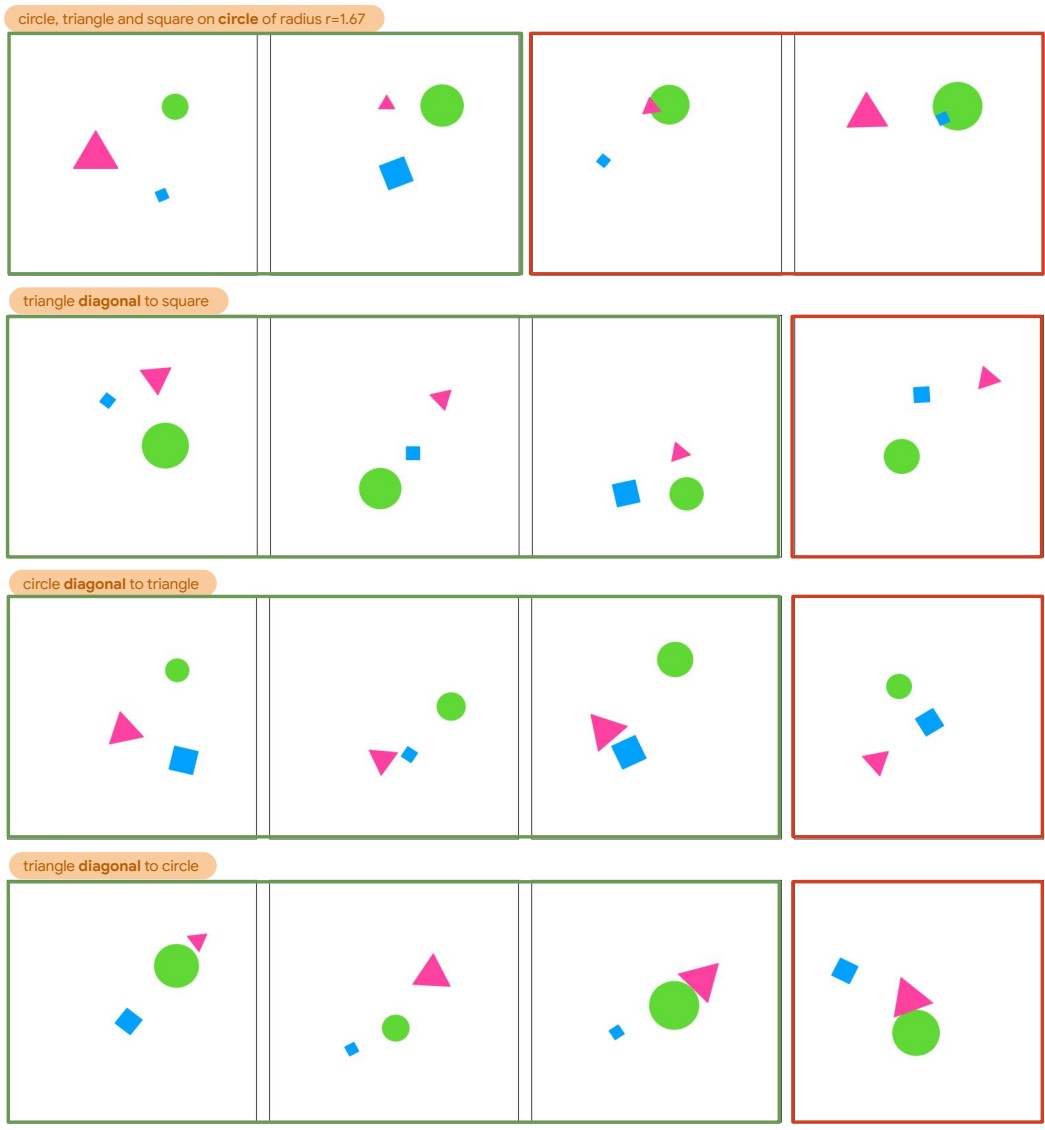

Figure 15: **New concepts that are not explicit training concept compositions.** We display successful (green frames) and unsuccessful (red frames) states generated by our model conditioned on learned concepts that are not explicit training concept compositions.

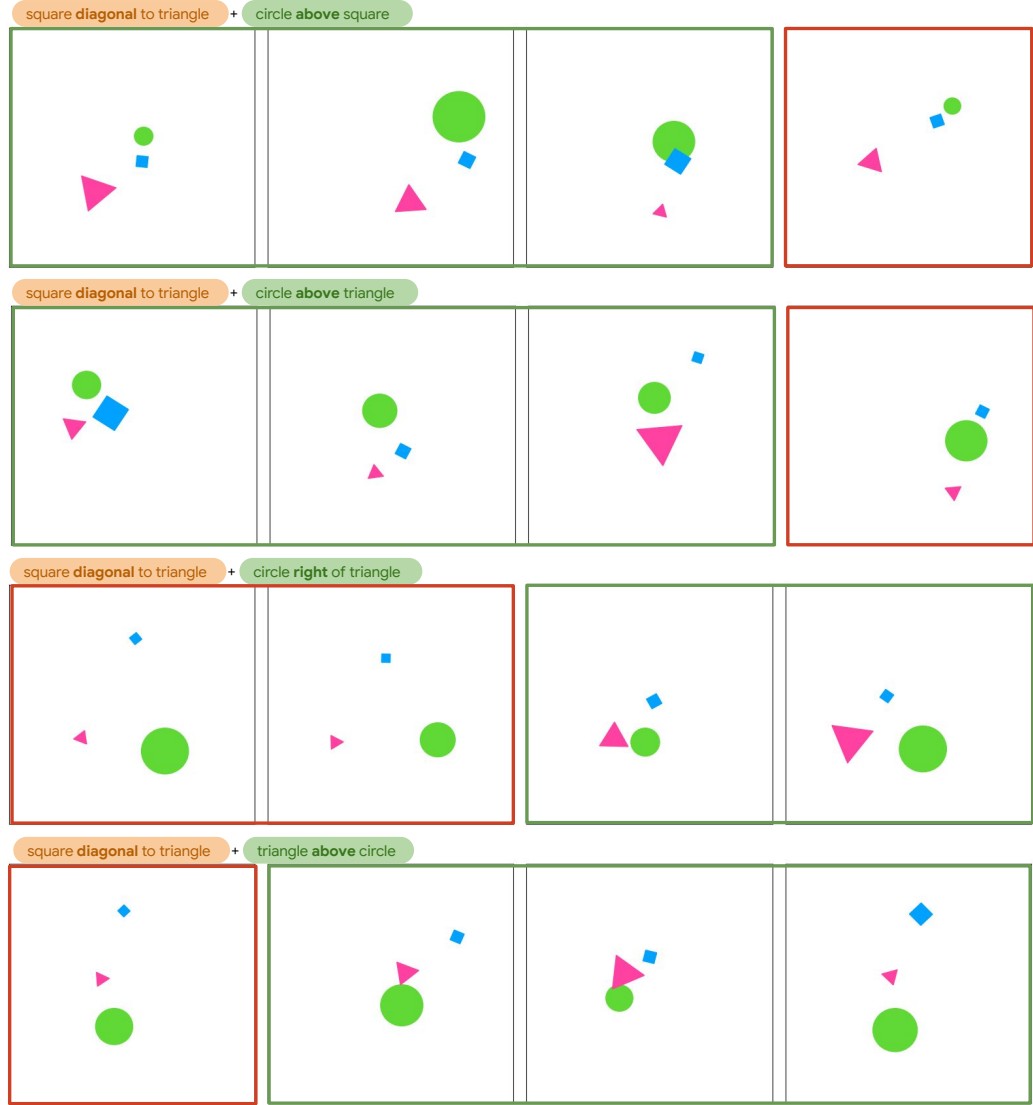

Figure 16: **New concept composed with training concepts.** We display successful (green frames) and unsuccessful (red frames) states generated by our model conditioned on a learned concept that is not an explicit training concept composition ('square diagonal to triangle') in composition with various training concepts.

Table 3: **Analysis of learned concepts.** For new concepts that are explicit training concept compositions, we evaluate what each learned component captures. For both concepts 1 and 2, we report accuracy with respect to the concept when trajectories are generated solely from one learned component. *e.g.* in AGENT (bottom), $42\%$ of the trajectories generated by component 1 and $48\%$ of the trajectories generated by component 2 target the red object.

| **Environment** | Concept 1 | Component 1 | Component 2 | Concept 2 | Component 1 | Component 2 |
|---|---|---|---|---|---|---|
| | $\triangle$R$\square$ | 0.42 | **0.62** | $\bigcirc$A$\square$ | **0.88** | 0.36 |
| | $\square$R$\triangle$ | 0.16 | **0.64** | $\bigcirc$A$\triangle$ | 0.08 | 0.08 |
| Object Rearrangement | $\bigcirc$R$\square$ | 0.1 | **0.48** | $\triangle$A$\square$ | **0.86** | 0.4 |
| | $\square$R$\bigcirc$ | 0.14 | **0.9** | $\triangle$A$\bigcirc$ | 0.02 | **0.6** |
| | $\bigcirc$R$\triangle$ | 0 | **0.92** | $\triangle$R$\square$ | **0.64** | 0.14 |
| | red | 0.42 | **0.48** | bowl | **0.66** | 0.56 |
| | yellow | **0.72** | 0.44 | bowl | 0.58 | **0.74** |
| AGENT | red | 0.42 | **0.76** | cube | 0.44 | **0.96** |
| | yellow | **0.4** | 0.32 | cube | **0.82** | 0.78 |
| | purple | **0.48** | 0.42 | cone | **0.56** | 0.36 |

'circle right of square $\wedge$ triangle above square', 'square right of circle $\wedge$ triangle above circle', 'line': 'circle right of triangle $\wedge$ triangle right of square'. There are five demonstrations of each novel composition.

- Five new concepts (new concept in Figure 6). 'circle': circle, triangle, and square lie in equal intervals on the circumference of a circle with radius 1.67 and center $\in [0,5]^2$, 'square diagonal to triangle', 'triangle diagonal to square', 'circle diagonal to triangle', 'triangle diagonal to circle'. A is considered diagonal to B if their centers lie on $f(x) = x$ and if A is 'above' and 'right of' B. There are five demonstrations of each novel concept.

- Composing a new concept, 'square diagonal to triangle', with five training concepts (new and training concept composition in Figure 6): 'circle right of square', 'circle above square', 'circle above triangle', 'circle right of triangle', 'triangle above circle'. When we learn weights for the new concept, the training concept is weighted by $\omega = 1$. Otherwise, both training and learned concepts are weighted by fixed $\omega$.

We verify that objects do not overlap and that no training relations unrelated to the specified new concepts exist in the demonstrations between objects.

**State space.** The state space is a 21-tuple describing three shapes (circle, triangle, and square), each represented by a 7-tuple: their center 2D position, size, angle, and one-hot shape type.

**Evaluation data and metrics.** For training, we generate 50 states conditioned on uniformly sampled training concept embeddings for single relations. For each concept composition and new concept, we report accuracy on 50 generated states conditioned on the learned concept. For new concepts composed with training concepts, we report accuracy based on 50 generated states for each training concept. Results in Figure 6 are reported for best learned $\omega$ and fixed $\omega \in \{1.2, 1.4, 1.6, 1.8\}$ – we learn two concepts with classifier-free guidance weight $\omega = 1.2$. We report accuracy based on a relaxed version of the data generation process: A is considered 'right of' B if $x_A > x_B$ and $|y_A - y_B| \leq 2 \cdot \max\{r_A, r_B\}$, A is considered 'above' B if $y_A > y_B$ and $|x_A - x_B| \leq 2 \cdot \max\{r_A, r_B\}$. A, B and C lie on a 'circle' if their centers form a circle of radius $r$ where $|r - 1.67| < 0.3$ and A is 'diagonal' to B if $x_A > x_B$, $y_A > y_B$ and they lie within a $2 \cdot \max\{r_A, r_B\}$ margin of $f(x) = x$.

## B.2 Goal-Oriented Navigation

**Training.** To collect demonstrations, we follow the data generation process in the AGENT benchmark environment [72], which provides a planner for navigation given the desired target. In the provided environment, the agent's initial position $a_{t=0}^p = (0, 0.102, -3.806)$, color (yellow) and shape (cone) are fixed. There are two objects: the target and a distractor. Each object has a color $o_1^c, o_2^c \in \{\text{red, yellow, purple, green}\}$, a shape $o_1^s, o_2^s \in \{\text{cone, sphere, bowl, cube}\}$ and a position $o_1^p \in [0, 1.66] \times \{0.102\} \times [-4.355, -3.257]$, $o_2^p \in [-1.66, 0] \times \{0.102\} \times [-4.355, -3.257]$.

The training dataset of $\sim 900$ samples consists of concepts 'go to red object' and 'go to yellow object' where $o_1^c \in \{\text{red}, \text{yellow}\}$, $o_2^c \in \{\text{red}, \text{yellow}\} \setminus \{o_1^c\}$ and $o_1^s, o_2^s \in \{\text{cone}, \text{sphere}\}$, and concepts 'go to bowl' and 'go to cube' where $o_1^s \in \{\text{bowl}, \text{cube}\}$, $o_2^s \in \{\text{bowl}, \text{cube}\} \setminus \{o_1^c\}$ and $o_1^c, o_2^c \in \{\text{purple}, \text{green}\}$.

**New Tasks.** The new scenarios include

- Five novel compositions of training target color and shape attributes (training composition in Figure 6). 'go to red bowl' ($o_i^c, o_j^c = \text{red}, o_i^s = \text{bowl}, o_j^s \in \{\text{cone}, \text{sphere}\}$ or $o_i^s, o_j^s = \text{bowl}, o_i^c = \text{red}, o_j^c \in \{\text{purple}, \text{green}\}$ where $i \in \{1, 2\}$ and $j = \{1, 2\} \setminus i$), 'go to yellow bowl' ($o_i^c, o_j^c = \text{yellow}, o_i^s = \text{bowl}, o_j^s \in \{\text{cone}, \text{sphere}\}$ or $o_i^s, o_j^s = \text{bowl}, o_i^c = \text{yellow}, o_j^c \in \{\text{purple}, \text{green}\}$), 'go to red cube' ($o_i^c, o_j^c = \text{red}, o_i^s = \text{cube}, o_j^s \in \{\text{cone}, \text{sphere}\}$ or $o_i^s, o_j^s = \text{cube}, o_i^c = \text{red}, o_j^c \in \{\text{purple}, \text{green}\}$), 'go to yellow cube' ($o_i^c, o_j^c = \text{yellow}, o_i^s = \text{cube}, o_j^s \in \{\text{cone}, \text{sphere}\}$ or $o_i^s, o_j^s = \text{cube}, o_i^c = \text{yellow}, o_j^c \in \{\text{purple}, \text{green}\}$), 'go to purple cone' ($o_i^c, o_j^c = \text{purple}, o_i^s = \text{cone}, o_j^s \in \{\text{bowl}, \text{cube}\}$ or $o_i^s, o_j^s = \text{cone}, o_i^c = \text{purple}, o_j^c \in \{\text{red}, \text{yellow}\}$). Note that in each scenario, the distractor object either has the same color or shape as the target, and combined with its other attribute (shape or color), it is within the training distribution (*i.e.*, red and yellow cones and spheres, and purple and green bowls and cubes). There are five demonstrations of each novel composition.

- Conditioning each novel composition concept on novel initial states sampled from the novel concept distribution (training composition new initial state in Figure 6).

**State space.** The **initial state space** we condition on is a 52-tuple based on the state space in Shu et al. [72] describing the agent and two objects. The agent is represented by its 3D position, quaternion $\in \mathbb{R}^4$, velocity $\in \mathbb{R}^3$, angular velocity $\in \mathbb{R}^3$, one-hot type (representing the agent and two objects), rgba color, and one-hot shape (representing the four possible shapes). Similarly, each object is represented by its 3D position, type, color, and shape. The **demonstration state space** is a 13-tuple representing the first 13 dimensions of the initial state space (agent position, quaternion, velocity, and angular velocity). Demonstrations $i \in [N]$ have horizons $H_i \leq 150$ and are padded to length 150 using the final state. During training, we sample the demonstrations to generate subtrajectories of length 128.

**Evaluation data and metrics.** For training, we generate 50 trajectories conditioned on uniformly sampled training concepts and initial states. For each new concept, we generate five trajectories conditioned on the learned concept and five initial states from the new concept demonstrations. To evaluate each concept composition on new initial states, we generate trajectories conditioned on the learned concept and 50 initial states sampled from the new concept distribution. Results in Figure 6 are reported for best learned $\omega$ and fixed $\omega \in \{1.2, 1.4, 1.6, 1.8\}$ – we learn two concepts with classifier-free guidance weight $\omega = 1.6$. We report accuracy based on whether the agent in the generated trajectory has made progress towards the desired target ($|a_{t=128}^p - o_{\text{target}}^p| < |a_{t=0}^p - o_{\text{target}}^p|$). Accuracy for closed loop evaluation (Figure 6) is reported based on optimally reaching target $o_{\text{target}}$ before a distractor $o_{\text{distractor}}$ ($|a_{t=128}^p - o_{\text{target}}^p| < 0.365 \wedge \forall t < 128 : |a_t^p - o_{\text{distractor}}^p| \geq 0.365$).

**Closed loop evaluation.** The action space $\mathcal{A} \subseteq \mathbb{R}^2$ represents forces applied to the agent. We take actions $a_t = \tau_{t+5} - s_t$ where $s_t$ is an observation in the environment, and $\tau_{t+5}$ is a future step in the open loop plan generated by $\mathcal{G}$ conditioned on observation $s_t$ and learned concept $\tilde{c}$. We execute actions in the environment until reaching the target or a maximum number of steps and report the evaluation metric described above. Closed loop evaluation is done in pybullet simulation [81] for efficiency.

## B.3 MoCap

**Training.** All human actions in the CMU Graphics Lab Motion Capture Database (http://mocap.cs.cmu.edu/) are included in the training set except three new scenarios. We further discard videos with less than 128 frames. The training set includes 2210 demonstrations.

**New Tasks.**    The new scenarios include

- Three new concepts. 'jumping jacks' (3 demonstrations), 'chop wood' (2), and 'breaststroke' (3).

- Conditioning each novel concept on novel initial states – the last states in the new concept demonstrations.

- Composing new learned concept 'jumping jacks' weighted by learned concept weights with three training concepts ('walk', 'jump' and 'march') weighted by $\omega = 1$, and conditioned on a training initial state from the training concepts.

**State space.**    The **initial state space** is a 42-tuple representing the 3D position of 14 joints. The original dataset contains 31 joints. Our version is adapted with Tanke et al. [82] to reduce the number of joints to 14 and to remove rotation and translation. The **demonstration state space** is the same with horizons $H_i, i \in [N]$. We subsample the original trajectories every 4 steps and further sample trajectories to generate subtrajectories of unified length 32 on which we train.

**Evaluation data and metrics.**    For each new concept, we generate trajectories conditioned on the learned concept and an initial state from the new concept demonstrations. For evaluating each new concept on new initial states, we generate trajectories conditioned on the learned concept and novel initial states from the new concept demonstrations that were not used as initial states during concept learning, specifically the last state in each demonstration. For new concepts composed with training concepts, we generate trajectories, each conditioned on the learned concept, a uniformly sampled training concept, and training initial state. Results are reported for best learned $\omega$ and fixed $\omega \in \{1.2, 1.4, 1.6, 1.8\}$ – we learn two concepts with classifier-free guidance weight $\omega = 1.8$.

**Human experiment.**    We show five humans (aged 23-28, four male) videos of training and new concepts with their natural language labels and ask whether each video depicts the concept and which depicts it best. The participants gave their consent, and the study was approved by an Institutional Review Board. We show three training concepts: 'march', 'run', and 'walk', and three new concepts: 'jumping jacks', 'chop wood', and 'breaststroke' on new initial states.

## B.4   Autonomous Driving

**Training.**    We use the following scenarios from the HighwayEnv driving simulation environment [75]. The **training** scenarios include four driving scenarios: 'highway', 'exit', 'merge', and 'intersection'. In 'highway', the objective is driving on a highway at high speed on the rightmost lanes while avoiding collisions. The highway has four lanes and 50 vehicles. The initial lane and position of all vehicles are sampled, as well as non-controlled vehicle speeds. An episode ends if the controlled vehicle crashes or a time limit is reached. In 'exit' the objective is to take a highway exit while driving on a four-lane highway with an exit lane and 20 vehicles. The controlled vehicle is rewarded for exiting at high speed and driving on the rightmost lanes while avoiding collisions. The initial position of all vehicles is sampled. An episode ends if the controlled vehicle crashes or a time limit is reached. In 'merge', the objective is driving on a highway with three lanes and a merging lane and three vehicles, one of them merging. The controlled vehicle is rewarded for driving at high speed while avoiding collisions and allowing another vehicle to merge into the highway. The lane, position, and speed are sampled for non-controlled vehicles. An episode ends if the controlled vehicle passes the merging lane or crashes. In 'intersection', the objective is making a left turn at an intersection with four two-way roads and 10 vehicles. The controlled vehicle is rewarded for crossing the intersection at high speed while staying on the road and avoiding collisions. The controlled vehicle's position is sampled, as well as the lane, position, and speed of other vehicles. An episode ends if the controlled vehicle completes crossing the intersection or crashes. Expert demonstrations were collected using a deterministic tree search planner provided in [83].

**New Task.**    The new scenario includes:

- A novel driving scenario: 'roundabout'. In this scenario, the objective is to take the second exit at a roundabout with four exits and four vehicles. The position and speed of non-controlled vehicles are sampled. An episode ends if the controlled vehicle crashes or a time

limit is reached. The novel concept is conditioned on novel initial states sampled from the new concept distribution. There are five demonstrations of this new concept.

**State space.** The **initial state space** we condition on is in $\mathbb{R}^{5 \times 7}$, the controlled vehicle and the four closest vehicles. Each vehicle is represented by a 7-tuple, including whether it is present on the road, its x and y positions, x and y velocities, and cosine and sine heading directions. The **demonstration state space** observations of the controlled vehicle in $\mathbb{R}^7$ for horizons $H_i$, $i \in [N]$. We sample trajectories to generate subtrajectories of length 8, which we train on.

**Evaluation data and metrics.** For evaluating the new concept on new initial states, we evaluate in closed loop on 50 initial states sampled from the new concept distribution. Task return, crash, and success rates are calculated based on the rewards described above. Note that the 'highway' scenario doesn't have a success score as there is no final state to reach in this scenario. Results are reported for best learned $\omega$ and fixed $\omega \in \{1.2, 1.4, 1.6, 1.8\}$ – for training, we report $\omega = 1.8$ and during new concept learning, we learn two concepts and their corresponding classifier-free guidance weights.

**Closed Loop Evaluation.** The action space is discrete: move to the left lane, stay idle, move to the right lane, drive faster, drive slower. We assume access to a planner $\mathcal{P}$ that given two states plans which action to take in the environment via access to simulation in the environment, $a_t = \mathcal{P}(\tau_{t+1}, s_t)$. The planner simulates the possible actions from observed state $s_t$ and randomly selects an action out of the ones closest to the model predicted next state $\tau_{t+1}$ that does not result in the controlled vehicle crashing. We execute actions until reaching a maximum number of steps or until the controlled vehicle crashes.

### B.5 Table-Top Manipulation

**Training.** We collect 214 expert demonstrations with a Franka Research 3 robot via teleop with a Spacemouse for four table-top manipulation tasks: 'pick green circle and place on book' (29 demonstrations), 'pick green circle and place on elevated white surface' (30), 'push green circle to orange triangle' (124), and 'push green circle to orange triangle around purple bowl' (31). Examples of these tasks are best viewed on our website.

**New task.** The new scenario includes pushing the green circle to the orange triangle on a book. We provide ten demonstrations of this task.

**State space.** The **initial state space** we condition on includes an overhead RGB image of the scene (Figure 10) and the robot's end effector pose, a 7-tuple of its 3D position and quaternion, and gripper state in $\mathbb{R}$. We verify that most states are fully observable. The **demonstration state space** includes observations of the end effector pose and gripper state in $\mathbb{R}^8$ for horizons $H_i$, $i \in [N]$. We sample trajectories to generate subtrajectories of length 32, which we train on.

**Evaluation data and metrics.** We evaluate 20 episodes for training 'push green circle to orange triangle' and new concept 'push green circle to orange triangle on book' on new initial states sampled from the concept distributions. An episode is successful if the green circle touches the orange triangle before a maximum horizon is reached. Results for training are reported for $\omega = 1.8$ and during new concept learning, we learn two concepts and their corresponding classifier-free guidance weights.

**Closed loop evaluation.** The action space is an 8-tuple of the end effector pose and gripper state, equivalent to the predictions of the model. Given predicted action $\tau_{t+1}$ and end effector pose $s_t$, we assume access to a planner that linearly interpolates between the current and next predicted pose. We make predictions and roll out the 32 predicted states in closed loop until it succeeds or executes a maximum number of steps $H = 160$.

## C Implementation Details

**Diffusion model.** We represent the noise model $\epsilon_\theta$ with an MLP for the object rearrangement domain and with a temporal U-Net for the AGENT, MoCap and Driving domains as implemented in Ajay et al. [17]. For Manipulation, we use a temporal U-Net where images are processed

by a pretrained resnet18 [84] that is finetuned with the model during training. We use the same hyperparameters as in Ajay et al. [17] except for the probability of removing conditioning information, $p$, which we set to $0.1$. In Table 4 we demonstrate the effect of choosing different classifier-free guidance weights $\omega$. We are overall better or comparable to the baselines.

Table 4: **Classifier-free guidance weight choice effect.** For Object Rearrangement and Goal-Oriented Navigation we report the accuracy and standard error of the mean for new concepts from new initial states, and for Driving, the success and crash rates. We compare the four baselines with our approach as reported in Figures 6, 9 and 13 and Section 5.2. We report results for our approach with all $\omega$ in our hyperparameter search and mark the reported $\omega$ in Figures 6, 9 and 13 in bold font.

| Domain | BC | VAE | In-Context | Language | $\omega$ | Ours |
|---|---|---|---|---|---|---|
| Object Rearrangement | $0.67 \pm 0.15$ | $0.08 \pm 0.05$ | - | $0.2 \pm 0.05$ | **1.2** | $0.82 \pm 0.09$ |
| | | | | | 1.4 | $0.8 \pm 0.1$ |
| | | | | | 1.6 | $0.66 \pm 0.08$ |
| | | | | | 1.8 | $0.61 \pm 0.06$ |
| | | | | | learned | $0.77 \pm 0.09$ |
| AGENT | $0.46 \pm 0.05$ | $0.6 \pm 0.06$ | $0.57 \pm 0.03$ | $0.63 \pm 0.07$ | 1.2 | $0.64 \pm 0.08$ |
| | | | | | 1.4 | $0.66 \pm 0.05$ |
| | | | | | **1.6** | $0.73 \pm 0.07$ |
| | | | | | 1.8 | $0.63 \pm 0.06$ |
| | | | | | learned | $0.67 \pm 0.05$ |
| Driving Success Rate | 4% | 24% | 0% | - | 1.2 | 18% |
| | | | | | 1.4 | 14% |
| | | | | | 1.6 | 4% |
| | | | | | 1.8 | 10% |
| | | | | | **learned** | 24% |
| Driving Crash Rate | 64% | 48% | 16% | - | 1.2 | 38% |
| | | | | | 1.4 | 40% |
| | | | | | 1.6 | 48% |
| | | | | | 1.8 | 40% |
| | | | | | **learned** | 32% |

**BC.** We implement behavior cloning (BC) [85]. The BC model is deterministic. In the Object Rearrangement domain, we only condition on concept $c$. To add stochasticity, when evaluating BC, we average results over 50 different seeds. We use an MLP with one hidden layer of size 512, ReLU activations, and AdamW [86] with learning rate $6 \cdot 10^{-4}$. In AGENT, at each step, the input to the model is the fully observable initial state (52-dim) and the condition, and the model learns to predict the next partially observable agent state (13-dim). Composing concepts is implemented by adding conditions.

**VAE.** We implement a conditional variational autoencoder model (CVAE) [76][2]. We use an MLP for the architecture with AdamW [86] and learning rate $6 \cdot 10^{-4}$. We sample 50 trajectories per concept and average the results. Composing concepts is implemented by adding conditions.

**In-Context.** For the In-Context learning baseline [8, 28], we use a transformer encoder-decoder architecture with 4 layers and 4 heads, where multiple demonstrations are passed into the encoder with a zero vector to separate consecutive demonstration trajectories. During training, there are five prompt demonstrations, during training evaluation, we provide one demonstration, and for new concepts, we provide two to five demonstrations, depending on the number of demonstrations provided in each domain. The decoder is passed a window of the previous states and predicts the next state. In AGENT and Driving, we use window size $K = 1$, and in MoCap $K = 2$. We simplify the learning objective to negative log-likelihood loss and convert the data to 20 bins for AGENT and Driving and 16 bins for MoCap. We further simplify the AGENT data by sampling the trajectories every eight states. We use learning rate $10^{-4}$ in all domains. We do not evaluate on Object Rearrangement since the horizon is one.

---

[2]based on the implementation in https://github.com/pytorch/examples/blob/main/vae/main.py

**Language.** The language instructions are as follows. For Object Rearrangement, training compositions are training concept descriptions composed with the word 'and'. New concepts are described as 'circle triangle and square in a circle of radius 1.67' and 'A right and above diagonally to B' where A, B∈ {'circle', 'triangle', 'square'}. New concepts composed with training concepts are composed with the word 'and'. AGENT concepts are described based on new concept attributes, *e.g.* 'go to red bowl'. In MoCap, descriptions are human actions such as 'jumping jacks'.

**Compute.** We run all simulated experiments on a single NVIDIA RTX A4000 machine. We evaluate our method on real-world table-top manipulation tasks using a Franka Research 3 robot with an overhead Realsense D435I RGB camera and an NVIDIA RTX 4090 machine. Concept learning can take approximately one to two hours. Since a new concept only has to be learned once to generate behavior, in the domains we presented, it is reasonable to learn a concept offline, and therefore, it is not prohibitive.

