# OpenReview forum: "Few-Shot Task Learning through Inverse Generative Modeling"
_NeurIPS.cc/2024/Conference — NeurIPS 2024 poster_

### Official Review · Reviewer_sy18 · 2024-06-20

**Soundness:** 3
**Presentation:** 4
**Contribution:** 3
**Rating:** 6
**Confidence:** 4

**Summary:**

The paper presents an approach to a new approach to few-shot learning. Specifically, the proposed method first pre-trains a conditional classifier-free guidance diffusion model on task concepts and their corresponding demonstrations. Few-shot learning is possible by inverting the diffusion model by optimizing for the task concepts.

**Strengths:**

1. The proposed method presents a possible solution to the few-shot learning task, although the method may have been applied in computer vision applications [1,2]
2. Experimental results suggest that by inverting a generative model to learn task concepts, the method works better than goal-conditioned or in-context learning-based baselines.
3. The method is clearly presented in the manuscript.

[1] Rinon Gal, Yuval Alaluf, Yuval Atzmon, Or Patashnik, Amit H Bermano, Gal Chechik, and Daniel Cohen-Or. An image is worth one word: Personalizing text-to-image generation using textual inversion. In International Conference on Learning Representations, 2023. 3
[2] Nan Liu, Yilun Du, Shuang Li, Joshua B Tenenbaum, and Antonio Torralba. Unsupervised compositional concepts discovery with text-to-image generative models. In International Conference on Computer Vision, 2023. 1, 3

**Weaknesses:**

1. For each new task, we need to find the concept vector that corresponds to the new task. This might introduce further complications at inference time compared to few-shot learning in LLM works, the language-conditioned and the in-context learning baseline.

**Questions:**

1. Are concept representations updated during training or are they fixed (as the T5 embeddings)?
2. How important is the classifier-free guidance weight? Does it require task-specific tuning?

**Limitations:**

Limitations are present in section 6.

---

> ### Author Rebuttal · Authors · 2024-08-05
>
> We thank the reviewer for the thoughtful review and positive feedback.
>
> >For each new task, we need to find the concept vector that corresponds to the new task. This might introduce further complications at inference time compared to few-shot learning in LLM works, the language-conditioned and the in-context learning baseline.
>
> The Language and In-Context baselines perform poorly in cases where the new concept is not an explicit composition of training tasks in natural language symbolic space, such as new object rearrangement concepts, new human motions and a new driving scenario. In these cases we demonstrate the benefit of providing demonstrations of a new task that require some learning but can then be used in various scenarios: (1) we keep the same concept latent representation, and change the initial state; (2) we combine the concept latent representation with other concept representations to form novel, compositional concepts. All of these can not be achieved without finding the concept representation. Since a new concept only has to be learned once to generate behavior, in the domains we presented, it is reasonable to learn a concept offline, and therefore not prohibitive.
>
> >Are concept representations updated during training or are they fixed (as the T5 embeddings)?
>
> Representations are fixed during training.
>
> >How important is the classifier-free guidance weight? Does it require task-specific tuning?
>
> In Table 4 (see the new pdf) we report results for various classifier-free guidance weights compared with the baselines. Overall for various choices of $\omega$ we are better or comparable to baselines. We find that learning the weights in all domains is a reasonable choice, and different domains may further benefit from tuning $\omega$ to a fixed value.

---

### Official Review · Reviewer_1rf8 · 2024-07-12

**Soundness:** 4
**Presentation:** 4
**Contribution:** 3
**Rating:** 5
**Confidence:** 3

**Summary:**

The paper addresses the challenge of learning the intents of an agent, such as its goals or motion style, from a few examples. The proposed approach, Few-Shot Task Learning Through Inverse Generative Modeling (FTL-IGM), leverages invertible neural generative models to learn new task concepts. The method involves pretraining a generative model on a set of basic concepts and their demonstrations, and then learning new concepts from a few demonstrations without updating the model weights. The approach is evaluated in four domains: object rearrangement, goal-oriented navigation, motion capture of human actions, and autonomous driving.

**Strengths:**

1. **Important Research Problem:** The paper addresses a critical problem in reinforcement learning—learning agent intents from limited data. This is a significant challenge with practical implications in various domains.

2. **Effective Visualization:** The paper provides effective visualizations, making the results and methodology easier to understand and interpret. The accompanying website further enhances the clarity and impact of the findings.

3. **Generalizability:** The approach demonstrates the ability to learn novel concepts and generate corresponding agent plans or motions in unseen environments and in composition with training concepts. This suggests a promising level of generalizability.

**Weaknesses:**

1. **Abstract Concept:** The concept of "task concept learning" is not rigorously defined and remains too abstract. The distinction between "concept" and "context" or "contextual" is unclear, which could lead to confusion about the precise nature and scope of the proposed methodology.

2. **Scalability Concerns:** Reinforcement learning environments vary widely in complexity and characteristics. The scalability of the pretrained model to diverse and complex real-world environments is not well addressed, raising concerns about its practical applicability.

**Questions:**

1. **Baseline Comparisons:** The paper compares the proposed method only against BC and VAE. Why were these baselines chosen, and how does the method compare to other relevant techniques in the field? Including a broader set of baselines would provide a more comprehensive evaluation of the approach's effectiveness.

**Limitations:**

The paper tackles an important problem in reinforcement learning with an innovative approach that leverages invertible neural generative models. While the visualizations and generalizability are strong points, the abstract nature of the core concept and scalability concerns in diverse environments need to be addressed. Additionally, expanding the set of baseline comparisons would strengthen the empirical validation of the method.

---

> ### Author Rebuttal · Authors · 2024-08-05
>
> We thank the reviewer for their review and suggestions for clarifying the paper.
>
> >The concept of "task concept learning" is not rigorously defined.
>
> In the Formulation section we define:
>
> - task concepts as latent representations in $\mathbb{R}^n$.
> - task concept learning as $argmax_{\tilde{c}}{\mathbb{E}}_{\tau\sim D _{\mathrm{new}}}[\log\mathcal{G} _{\theta}(\tau|\tilde{c},s_0)]$.
>
> We state this in the 4th paragraph of the Introduction as well: *“To learn new tasks from a limited number of demonstrations, we then formulate few-shot task learning as an inverse generative modeling problem, where we find the latent task description, which we refer to as a concept, which maximizes the likelihood of generating the demonstrations.”*
>
> Intuitively, task concept learning means inferring the intent of a demonstration and can be formally defined in various ways such as policies, rewards or trajectories, as we discuss in the 2nd paragraph of the Introduction. We will edit the 3rd paragraph of the Introduction to make clear what we mean by task concept learning earlier on in the paper.
>
> >The distinction between "concept" and "context" or "contextual" is unclear.
>
> Concept refers to a latent task representation that we learn. Context is **only** used in relation to *In-Context* [1], a baseline we compare with. We do not use context as a technical term in our formulation. We will change "in context" appearances in the paper to "in-context" for clarity.
>
> >The scalability of the pretrained model to diverse and complex real-world environments is not well addressed, raising concerns about its practical applicability
>
> We evaluate our method's capability to learn a novel concept for **real-world table-top manipulation with a Franka Research 3 robot**. We train on a suite of tasks (including table-top pick-and-place onto elevated surfaces, and table-top pushing scenarios) and learn a new task that requires high precision (pushing on an elevated surface) from ten demonstrations, **see Figure 15 in the new pdf**. We evaluate in closed loop and achieve success rate of 0.9 on training pushing, **0.55 success on the learned new concept**, elevated pushing, and surpass a **baseline** that conditioned on the training pushing concept, **achieves 0.15 success rate** in the elevated pushing setup. **For more details please see the general response**.
>
> >The paper compares the proposed method only against BC and VAE. Why were these baselines chosen, and how does the method compare to other relevant techniques in the field?
>
> The paper compares against **four baselines**. As described in the Experiments section, **in addition to BC and VAE, we compare with In-Context and Language baselines**. The **In-Context** baseline [1] (**Figures 5,8 & 10 and https://sites.google.com/view/FTL-IGM/home**) represents tasks with demonstrations and generates behavior conditioned on those demonstrations. For new concepts, behavior is generated in a zero-shot manner by conditioning on the new demonstrations. For the **Language** baseline (**Section 5.2, *Conditioning on Language Descriptions of New Concepts*** **and https://sites.google.com/view/FTL-IGM/home**), we do not assume access to new concept demonstrations and instead represent the new task with a T5 embedding of its language description (as we do for labeled training concepts). We input this representation to our model to zero-shot produce new behavior.
>
> Our work is in the field of learning from demonstrations. As described in the Related Work section, the main approaches in this field are BC, IRL, Inverse Planning and In-Context learning. Specifically, we focus on few-shot task representation learning from demonstrations. Each baseline highlights a different aspect of our approach.
>
> - The **BC** baseline highlights the generative aspect of our approach. BC is **autoregressive** (predicts one state at a time given past states and a latent task representation) whereas our approach is generative (predicts 𝐻 future states from an initial state and latent task representation).
> - Similar to our approach, VAEs are generative models. The **VAE** baseline highlights our choice of representing tasks as T5 embeddings during training. VAE does not utilize these representations during training and instead **learns latent task representations** from demonstrations in a self-supervised manner.
> - The **In-Context** baseline [1] is **autoregressive** and **represents tasks as demonstrations**.
> - The **Language** baseline utilizes our **generative** model in an **In-Context** fashion and highlights the need for demonstrating new concepts.
>
> IRL is not relevant for few-shot learning as it learns only one task representation (reward or policy) at a time and assumes access to taking actions in the environment during training. Inverse Planning is not suitable for our case as it assumes knowledge about the task space. We also discuss why fine-tuning task conditioned BC is not possible — we assume no access to new concept task representations, only to their demonstrations.
>
> [1] Prompting decision transformer for few-shot policy generalization. Xu et al. 2022.

---

> > ### Comment · Reviewer_1rf8 · 2024-08-08
> > **Raising the score**
> >
> > The responses address my concerns. I thus raise the score.

---

### Official Review · Reviewer_gXQZ · 2024-07-12

**Soundness:** 4
**Presentation:** 3
**Contribution:** 4
**Rating:** 7
**Confidence:** 5

**Summary:**

The paper focuses on learning concepts that describe behaviors seen in state-based trajectories, such as mocap data or simplified autonomous driving simulation. The proposed approach uses a generative model to predict state trajectories based on concepts annotated with natural language. Next, new trajectories without annotated concepts are provided to the model. The method uses gradient descent to invert the network and optimize new concept embeddings for the novel trajectories. Then, the learned concepts embeddings are shown to generate correct trajectories in novel starting states. Moreover, the authors demonstrate composable concept embeddings.

**Strengths:**

1. The authors propose a novel method that learns concepts by optimizing the input concept embedding to a pre-trained generative model.

2. The method also allows for composable concepts based on prior work on composable diffusion models.

3. The method is evaluated in a diverse set of environments, the baselines the authors use are reasonable.

**Weaknesses:**

1. Section 4.2 could use more explanation. In particular, is there something specific about the diffusion model that makes it “invertible”? As far as I understand, optimizing with respect to the input using gradient descent is possible with all neural networks. Usually, the community uses the term “invertible networks” (this term is used in the paper) to mean a neural network F for which computing F^{-1} is easy [1, 2]. Moreover, Equation 2 could be explained better so that the paper is more self-contained.

2. Only state-based domains are used. It is unclear if this approach learns meaningful concepts when used with an image or a video generative model.

3. The exposition could be improved. The methods section is very short; a background section could prepare readers to understand it better. Moreover, the related work section would benefit from further discussion of generative models and composable representations.

References:

[1] Invertible Residual Networks. Behrmann et al. 2018.

[2] Analyzing inverse problems with invertible neural networks. Ardizzone et al. 2018.

**Questions:**

1. Is unconstrained gradient descent with respect to the input concept embedding the right way to learn concepts? For example, in the “deep dream” literature [3], gradient descent is used to create an input image that maximizes a particular neuron in the network. In this case, normal gradient descent does not work very well because it introduces only high frequency patterns in the image. There are various approaches that add noise to the gradients for smoothing, etc.

[3] https://research.google/blog/inceptionism-going-deeper-into-neural-networks/

**Limitations:**

The limitations of the method are addressed.

---

> ### Author Rebuttal · Authors · 2024-08-06
>
> We thank the reviewer for the thoughtful review and positive feedback.
>
> >is there something specific about the diffusion model that makes it “invertible”?
>
> No, we state in Limitations that *“our framework is general for any parameterized generative model”*. We choose to implement our framework with a diffusion model given its success on interpolation and compositionality, both in learning compositions of concepts and in generating compositions of concepts. We describe these properties in the Introduction.
>
> >the community uses the term “invertible networks”
>
> We refer to inverting the generative model [4-6] as opposed to inverting the model.
>
> >Equation 2 could be explained better so that the paper is more self-contained.
> The methods section is very short; a background section could prepare readers to understand it better.
>
> We will extend the method section to include the formulation and objectives of diffusion models [7], conditional diffusion models [8, 9], classifier-free guidance [10], and concept learning in computer vision [11], including for multiple visual concepts [12].
>
> >It is unclear if this approach learns meaningful concepts when used with an image or a video generative model.
>
> We evaluate our method's capability to learn a novel concept for **real-world table-top manipulation with a Franka Research 3 robot conditioned on RGB images**. We train on a suite of tasks (table-top pick-and-place onto elevated surfaces, and table-top pushing scenarios) and learn a new task that requires high precision (pushing on an elevated surface) from ten demonstrations, **see Figure 15 in new pdf**. We evaluate in closed loop and achieve success rate of 0.9 on training pushing, **0.55 success on the learned new concept**, elevated pushing, and surpass a **baseline** that conditioned on the training pushing concept, achieves **0.15 success rate** in the elevated pushing setup. For more details please see general response.
>
> >related work section would benefit from further discussion of generative models and composable representations.
>
> We will add the following discussion to the related work:
>
> Generative Models in Decision Making. The success of diffusion policy in predicting sequences of future actions has led to 3D extensions [13], and combined with ongoing  robotic data collection efforts [14] and advanced vision and language models, has led to vision-language-action generative models [15,16].
>
> Composable representations. There has been work on obtaining composable data representations. $\beta$-VAE [17] learns unsupervised disentangled representations for images. MONet [18] and IODINE [19] decompose visual scenes via segmentation masks and COMET [20] and [12] via energy functions. There is also work on composing representations to generate data with composed concepts. Generative models can be composed together to generate visual concepts [21-26] and robotic skills [27]. The generative process can also be altered to generate compositions of visual [28-31] and molecular [32] concepts. We aim to obtain task concepts and generate them in composition with other task concepts.
>
> >Is unconstrained gradient descent with respect to the input concept embedding the right way to learn concepts?
>
> Concept learning is not unconstrained. Unlike in deep dream, in this work we do not maximize a particular neuron, instead we optimize the input concept for generating the demonstrated new concept, therefore this optimization is supervised.
>
>
> [4] Action understanding as inverse planning. Baker et al. 2009
>
> [5] Rational quantitative attribution of beliefs, desires and percepts in human mentalizing. Baker et al. 2017
>
> [6] Online bayesian goal inference for boundedly rational planning agents. Zhi-Xuan et al. 2020
>
> [7] Denoising Diffusion Probabilistic Models, Ho et al. 2020
>
> [8] Diffusion Models Beat GANs on Image Synthesis, Dhariwal & Nichol 2021
>
> [9] Hierarchical Text-Conditional Image Generation with CLIP Latents, Ramesh et al., 2022
>
> [10] Classifier-Free Diffusion Guidance, Ho & Salimans, 2022
>
> [11] An Image is Worth One Word: Personalizing Text-to-Image Generation using Textual Inversion, Gal et al. 2023
>
> [12] Unsupervised Compositional Concepts Discovery with Text-to-Image Generative Models. Liu et al. 2023
>
> [13] 3D Diffusion Policy. Ze et al. 2024
>
> [14] Open X-Embodiment: Robotic Learning Datasets and RT-X Models. 2024
>
> [15] 3D-VLA: A 3D Vision-Language-Action Generative World Model. Zhen et al. 2024
>
> [16] Octo: An Open-Source Generalist Robot Policy. Ghosh et al. 2024
>
> [17] $\beta$-VAE: Learning basic visual concepts with a constrained variational framework. Higgins et al. 2017
>
> [18] MONet: Unsupervised Scene Decomposition and Representation. Burgess et al. 2019
>
> [19] Multi-Object Representation Learning with Iterative Variational Inference. Greff et al. 2020
>
> [20] Unsupervised Learning of Compositional Energy Concepts. Du et al. 2021
>
> [21] Learning to Compose Visual Relations. Liu et al. 2021
>
> [22] Compositional Visual Generation with Composable Diffusion Models. Liu et al. 2023
>
> [23] Controllable and Compositional Generation with Latent-Space Energy-Based Models. Nie et al. 2021
>
> [24] Reduce, Reuse, Recycle: Compositional Generation with Energy-Based Diffusion Models and MCMC. Du et al. 2023
>
> [25] Concept Algebra for (Score-Based) Text-Controlled Generative Models. Wang et al
>
> [26] Compositional Visual Generation with Energy Based Models. Du et al. 2020
>
> [27] Is Conditional Generative Modeling All You Need For Decision-Making? Ajay et al. 2023
>
> [28] Training-Free Structured Diffusion Guidance For Compositional Text-To-Image Synthesis. Feng et al. 2023
>
> [29] Exploring Compositional Visual Generation with Latent Classifier Guidance. Shi et al. 2023
>
> [30] Attribute-Centric Compositional Text-to-Image Generation. Cong et al. 2023
>
> [31] Composer: Creative and Controllable Image Synthesis with Composable Conditions. Huang et al. 2023
>
> [32] Compositional Sculpting Of Iterative Generative Processes. Garipov et al. 2023

---

### Author Rebuttal · Authors · 2024-08-05

We thank the reviewers for their feedback and for acknowledging that learning agent intent from limited data is an important problem, and that our method is evaluated in a diverse set of environments, surpassing baseline performance and demonstrates generalizability, including compositionality. Moreover, we thank the reviewers for acknowledging that the paper is clear and provides effective visualizations.

We describe additional experiments (please see new pdf) here to address reviewers’ comments, and address individual comments in response to each reviewer below.

# **Real-World Experiments**

We evaluate our method's capability to learn a novel concept for **real-world table-top manipulation with a Franka Research 3 robot**. We **train** on a suite of tasks (including **table-top pick-and-place onto elevated surfaces, and table-top pushing scenarios**) and learn a **new task** that requires high precision (**pushing on an elevated surface**) from ten demonstrations, see **Figure 15** in new pdf. We **evaluate in closed loop** and achieve success rate of **0.9** on **training** pushing, **0.55** success on the **learned new concept**, elevated pushing, and surpass a **baseline** that conditioned on the training pushing concept, achieves **0.15** success rate in the elevated pushing setup.

Specifically, we collect 214 expert demonstrations with a Franka Research 3 robot via teleop with a Spacemouse for four table-top manipulation tasks: pick green circle and place on book (29 demonstrations), pick green circle and place on elevated white surface (30), push green circle to orange triangle (124) and push green circle to orange triangle around purple bowl (31). The new concept includes pushing the green circle to the orange triangle on a book. We provide ten demonstrations of this task. Demonstrations have horizons $H_i,\,i\in[N]$ which we split into subtrajectories of length 32. Our generative **model learns to predict the next 32 states**, each composed of the **end effector pose**, a 7-tuple of its 3D position and quaternion, **and gripper state** $\in\mathbb{R}^8$, **conditioned on** an overhead **RGB image** of the scene, and the **robot's current end effector pose and gripper state**. We verify that most states are fully observable.

For closed loop evaluation, given the predicted state $\tau_{t+1}$ and current end effector pose $s_t$, we assume access to a planner that linearly interpolates between them. We repeatedly make predictions and roll out the 32 predicted states in closed loop until success or upon executing a maximum number of steps $H=160$. The success rate we report above is for 20 episodes with new initial states sampled from the training and new concept distributions. An episode is successful if the green circle touches the orange triangle before a maximum horizon is reached. Results for training are reported with $\omega=1.8$ and during new concept learning, we learn two concepts and their corresponding classifier-free guidance weights. The Diffusion network architecture is a temporal U-Net where images are processed by a pretrained resnet18 [1] that is finetuned with the model during training. In our manipulation experiments we use a Realsense D435I RGB camera and an NVIDIA RTX 4090 machine.

# **Classifier-Free Guidance Parameter**

In Table 4 (see new pdf) we report results for various classifier-free guidance weights compared with the baselines in the Object Rearrangement, Goal-Oriented Navigation and Driving domains. For Object Rearrangement and Goal-Oriented Navigation we report the accuracy and standard error of the mean for new concepts from new initial states, and for Driving, the success and crash rates. We compare the four baselines with our approach as reported in Figures 5, 8 and 10 and section 5.2. We report results for our approach with all $\omega$ in our hyperparameter search, and mark the reported $\omega$ in Figures 5, 8 and 10 in bold font. Overall for various choices of $\omega$ we are better or comparable to the baselines. **We find that learning the weights along with concepts in all domains is a reasonable choice, and different domains may further benefit from tuning $\omega$ to a fixed value.**

[1] Deep residual learning for image recognition. He, et al. 2016.

---

### Decision · Program_Chairs · 2024-09-25

**Decision:**

Accept (poster)

**Comment:**

The submission introduces a novel few-shot learning framework for understanding agent intents such as goals or motion styles from few examples, for multiple application domains (e.g., motion capture and autonomous driving). The proposed method leverages a pre-trained invertible neural generative model, initially trained on a variety of basic tasks and their demonstrations. By inverting the diffusion model, the method can obtain new concept embedding, which then can be used to generate the new trajectories (given any new state). The authors validated the method in diverse tasks including object rearrangement, goal-oriented navigation, and human action simulations.

All reviewers agreed that the paper showed good results and the method is timely interesting to the community. So, I recommend accept. There are a few suggestions, such as improving the writing of the method section and including more baselines. Please add them to your final copy.